# Adapting a Participatory Group Programme for Caregivers of Children with Complex Neurodisability from Low-, Middle-Income Countries to a High-Income Setting: Moving from “Baby Ubuntu” to “Encompass”

**DOI:** 10.3390/ijerph22071144

**Published:** 2025-07-18

**Authors:** Kirsten Prest, Kirsten Barnicot, Catherine Hurt, Frances Badenhorst, Aleksandra Borek, Melanie Whyte, Phillip Harniess, Alea Jannath, Rachel Lassman, Christopher Morris, Rachel Osbourne, Tracey Smythe, Cally J. Tann, Keely Thomas, Emma Wilson, Angela Harden, Michelle Heys

**Affiliations:** 1School of Health and Medical Sciences, City St George’s, University of London, London EC1V 0HB, UK; kirsten.barnicot@citystgeorges.ac.uk (K.B.); catherine.hurt.1@citystgeorges.ac.uk (C.H.); angela.harden@citystgeorges.ac.uk (A.H.); 2Specialist Children’s and Young People’s Services, Newham, East London NHS Foundation Trust, London E15 4PT, UK; m.heys@ucl.ac.uk; 3Occupational Therapy, Barts Health NHS Trust, London E1 4DG, UK; frances.badenhorst@nhs.net; 4Institute of Psychology, SWPS University, 53-238 Wroclaw, Poland; aleksandra.borek@swps.edu.pl; 5Lived Experience Research Partner, London EC1V 0HB, UK; 6Faculty of Health and Life Sciences, Northumbria University, Newcastle upon Tyne NE7 7XA, UK; phill.harniess@northumbria.ac.uk; 7Peninsula Childhood Disability Research Unit (PenCRU), St. Luke’s Campus, University of Exeter Medical School, Exeter EX1 2HZ, UK; christopher.morris@exeter.ac.uk; 8London School of Hygiene and Tropical Medicine, London WC1E 7HT, UK; rachel.lassman@lshtm.ac.uk (R.L.); cally.tann@lshtm.ac.uk (C.J.T.); 9Department of Health and Rehabilitation Sciences, Division of Physiotherapy, Stellenbosch University, Cape Town 7602, South Africa; 10Disability Research Group, MRC/UVRI & LSHTM Uganda Research Unit, Entebbe P.O. Box 49, Uganda; 11Neonatal Medicine, University College London Hospitals NHS Trust, London NW1 2BU, UK; 12Great Ormond Street Institute for Child Health (ICH), University College London, London WC1N 1EH, UK; e.k.wilson@ucl.ac.uk

**Keywords:** complex neurodisability, cerebral palsy, child disability, caregivers, family-centred care, community-based interventions, support groups, intervention adaptation, peer support, participatory approach

## Abstract

The “Baby Ubuntu” programme is a well-established, low-cost, community-based intervention to support caregivers of children with complex neurodisability, like cerebral palsy, in low- and middle-income country (LMIC) contexts. This process-focused paper describes our utilisation of the ADAPT guidance to adapt “Baby Ubuntu” for use in ethnically and linguistically diverse, and economically deprived urban boroughs in the United Kingdom (UK). The process was guided by an adaptation team, including parents with lived experience, who explored the rationale for the intervention from local perspectives and its fit for this UK community. Through qualitative interviews and co-creation strategies, the perspectives of caregivers and healthcare professionals substantially contributed to the “Encompass” programme theory, drafting the content, and planning the delivery. Ten modules were co-produced with various topics, based on the “Baby Ubuntu” modules, to be co-facilitated by a parent with lived experience and a healthcare professional. The programme is participatory, allowing caregivers to share information, problem solve, and form supportive peer networks. The “Encompass” programme is an example of a “decolonised healthcare innovation”, as it aims to transfer knowledge and solutions developed in low- and middle-income countries to a high-income context like the UK. Piloting of the new programme is underway.

## 1. Introduction

Cerebral palsy (CP) refers to a group of disorders caused by an injury to the developing brain that affects movement, posture, and muscle tone [1]. It is the most common cause of a physical childhood-onset disability, affecting 1 in 400 births in the United Kingdom (UK) [2]. These difficulties result in functional limitations as well as participation and activity restrictions [3]. Children with CP often present with numerous additional conditions, which require a multi-disciplinary team approach [4]. As a result, their parents and primary caregivers are required to manage the multiple services and professionals involved, which can be overwhelming and result in feelings of stress, guilt, and inadequacy [5]. A disproportionate number of healthcare services are involved in their care, along with services from social care, education, and charity sectors [6,7]. The high frequency of medical appointments among children with CP is not only attributed to their co-morbidities, but also to inadequately coordinated and integrated health services [8], an issue that becomes more pronounced during adolescence and adulthood [9]. In this paper, we use the term “caregivers” to inclusively refer to parents, relatives, and others who provide primary care for the child.

Receiving a diagnosis like CP is a life-altering moment for caregivers, as they face multiple challenges in caring for their child, their family, and themselves [5]. Simultaneously, they are learning to cope with the grief of losing their envisioned future, accepting their child’s disability, dealing with their own emotions, and juggling a deluge of appointments that arrive on their doorstep [10]. Caregivers of children with complex neurodisability, such as CP, consistently present with adverse health outcomes such as increased levels of depression, anxiety, and musculoskeletal disorders [11,12,13]. Factors contributing to these health challenges include: a lack of understanding from friends and family, feelings of isolation, a lack of knowledge, financial worries, the physical challenges of moving and handling their child and dealing with the continuous cycle of challenges [11,14,15].

The difficulties faced by caregivers of children with CP are further exacerbated in the ethnically and linguistically diverse, and economically deprived urban areas of East London in the UK, where this study is based. These areas experience significantly higher rates of children living in poverty, along with increased mental and physical health issues, and higher service use compared to the rest of the UK [16,17,18]. They also have reduced health literacy [19] and a low proportion of first-language English speakers [20].

### 1.1. Family-Centred Care

Family-centred care, the gold standard approach to delivering healthcare for children with complex conditions, requires working together in partnership with caregivers to support the unique and individual needs of each child and family [21,22]. Family-centred care describes an approach to decision-making between the family and health professional with principles including information sharing, respecting and honouring differences, partnership and collaboration, negotiation, and care within the family and community context [23]. However, a gap remains in our understanding and ability to deliver family-centred care, particularly in diverse urban settings with increased levels of child poverty [18], low rates of health literacy [19], and reduced wellbeing of the population [16,17]. These challenges present additional barriers to accessing health and social care services and increase the health inequalities experienced by families of children with complex neurodisability. Our recent qualitative study conducted in East London (UK) highlighted the mental health challenges experienced by caregivers of children with complex neurodisabilities, the lack of provision of knowledge about their child’s condition, and the disjointed care that they experience while attempting to juggle multiple services, each with its own jargon and language [24]. The “Ubuntu Hub” has been highlighted as a source of evidence-based, low-cost programmes to improve the provision of appropriate family-centred support for families of children with complex neurodisability in the UK [25].

### 1.2. The Ubuntu Hub

The Ubuntu Hub is a non-profit shared learning and research hub providing participatory programmes of care and support for children with developmental disabilities and their caregivers, rooted in the African philosophy of ‘Ubuntu,’ which champions community and shared humanity and based on the principles of adult learning theory [26] and family-centred care. Programme goals are to promote inclusion and participation of children with disabilities, support developmental progress, understand and respond to the lived experience of caregivers, and promote caregiver agency through accessible information, skills building, and peer support. Originally developed from “Getting to Know Cerebral Palsy”, the Hub now includes several programmes for use in a wide range of contexts. These include “Baby Ubuntu” for children 0–3 years [27], Ubuntu Kids for children 2–12 years [28], Juntos [29,30] for children with congenital Zika syndrome, and Obuntu Bulamu for inclusive primary education [31].

Mixed methods evaluations of the Ubuntu Hub programmes have demonstrated improvements in caregiver wellbeing and confidence in caring for their child, increased peer support, improved child development, behaviour, and wellbeing, and greater understanding and attitudes towards children with disabilities [28,29,30,31]. The “Baby Ubuntu” feasibility trial in Uganda showed the programme to be low-cost, feasible, and acceptable from the perspective of the families and providers, in both urban and rural settings [27] and pre-/post-evaluations have reported a 20–25% improvement in family impact quality of life [32]. A single-blind, effectiveness implementation–hybrid (type II) cluster randomised trial of the “Baby Ubuntu” programme integrated with government health services in Rwanda is currently underway [33]. To date, the Ubuntu programmes are being implemented across 13 countries, including East and West Africa, South America, and more recently in India and Guatemala; however, no formal adaptation for high-income countries (HICs) as yet exists.

### 1.3. Frugal Innovations in the UK

Although there has been a recent steer towards considering low-cost or ‘frugal’ health innovations for HICs like the UK and US [34,35], there are barriers to this in practice. There is an unconscious bias that the ‘correct’ direction of knowledge transfer flows from HICs to LMICs [36]. Furthermore, global research and innovation are skewed towards the West [37]. Even the term ‘reverse innovation’, which describes the adoption of innovations developed in LMICs for HICs, reiterates the direction that the concept is attempting to dismantle [38]. A low-cost innovation that solves a problem for a community might be just as likely to be needed in a context, such as the UK’s National Health Service (NHS), which is currently facing increased pressures and a workforce crisis [39]. The NHS has seen an increase in the adoption of frugal technologies, particularly in response to the COVID-19 pandemic; however, there has been encouragement to consider these innovations beyond times of crises [40]. A higher-cost solution is unlikely to be feasible, acceptable, or solve challenges in the under-resourced and pressured context of the NHS. The bi-directional flow of innovations between settings and the adoption of frugal innovations by the NHS are increasingly important when solving implementation problems [36], such as the challenges documented above in providing family-centred care for children with complex neurodisability.

This paper reports on the process used to adapt a low-cost innovation (the Ubuntu programme) developed in LMICs into ethnically and linguistically diverse, and economically deprived areas within a HIC (the UK).

The adaptation process aimed to address the following questions:What should the content of the new programme be? What should the adapted content look like?How should the adapted programme be delivered in the new context?How can the new programme best reach diverse and underserved populations in the London boroughs?

This paper describes the steps taken to adapt the “Baby Ubuntu” caregiver group support programme. The new programme (re-named “Encompass”) and its structure are described along with the logic model describing how the programme is expected to work. Please refer to Figure 1 to understand how this publication fits into the wider programme of work.

## 2. Methods

The programme was adapted using the ADAPT framework [42] with results presented based on recommended reporting items.

The adaptation process involved:The formation of an adaptation team.Exploring the rationale for the intervention from local perspectives.Exploring the intervention fit of “Baby Ubuntu” for East London, UK.Gathering recommendations from local caregivers and healthcare professionals on the content and delivery of the intervention and how to reach diverse populations.Drafting the adapted programme manual and programme theory.

Further information about how the ADAPT process items were operationalised may be found in Section A.1. Although implementation data is not yet available, a pilot and feasibility study [41] is currently underway with results to be shared in a future publication.

### 2.1. Adaptation Team

The adaptation team was formed at the beginning of the project. It consisted of the core research group (PhD student K.P. and four supervisors A.H., M.H., K.B., C.H.) as well as caregivers with lived experience (A.J., K.T., M.C., R.O.). There were also health professionals and academic researchers who were involved in the original development of the “Baby Ubuntu” programme and subsequent implementation and adaptations in LMICs (C.J.T., R.L., T.S.). Finally, there were professionals on the team with expertise in relation to the clinical population, the development, and evaluation of complex interventions, and the NHS/UK context (C.M., E.W., P.H., A.B., F.B.).

The adaptation team met every 6 months to explore key uncertainties, share local perspectives, develop the programme theory, and make decisions about the programme manual and delivery plan.

The parents with lived experience included four mothers (A.J., K.T., M.W., R.O.) who were recruited through two community child health centres and by word of mouth from other parents. Attempts were made to attract a diverse group of caregivers who could represent a variety of perspectives. No fathers put themselves forward. Although the mothers formed a diverse group of ages, ethnicities, and cultures, they shared resemblances in their journeys of being mothers of adolescents/young people with complex neurodisability. It was made clear from the beginning that this would not be a one-off project and that the group could continue to meet formally or informally after this study ended. The Involvement Matrix tool [43] was utilised at initial meetings as a way of promoting a more equitable power structure. Their contributions and expertise are acknowledged through co-authorship [44]. The adaptation team members who were involved in the original development of the “Ubuntu” programme (C.J.T., R.L., T.S.) provided answers to questions around uncertainties (e.g., characteristics of the venue required), offered useful tips for the adaptation process (e.g., recording essential decisions made with regard to the content development and delivery plan with clear justifications), and stimulated discussion around context (e.g., conceptualising Uganda’s health system in relation to non-profit organisations, in comparison to the UK’s government-funded health system which better reflects implementation of “Baby Ubuntu” in Rwanda).

These parents with lived experience initially met separately with K.P. to build their confidence in becoming co-creators [45] and to ensure their voices were heard. Once they felt more confident, they joined the larger adaptation group meetings to share their perspectives with the wider team. The professionals in the adaptation team were experienced in partnering with families in research and were, therefore, conscious of the power imbalances. Every attempt was made to create an inclusive culture within these meetings that valued expertise through experience as much as professional knowledge. K.P. facilitated these meetings, and efforts were made to ensure that every person had an opportunity to be heard. Debrief sessions also took place with the parent partners after the meetings, if required.

### 2.2. Local Perspectives

We conducted a qualitative interview study to explore local perspectives relating to processes 2–4 in the ADAPT framework (see above). These included exploring (2) the rationale for the intervention, (3) the intervention fit, and (4) local recommendations for the new programme manual and delivery plan. Figure 1 shows the results from the qualitative study that have been reported in this paper.

Twelve caregiver participants were recruited from an inner-city area in East London, UK, that is ethnically, culturally, and linguistically diverse, with high levels of poverty. Most were mothers (*n* = 10) with ages ranging from 31 to 42 years. There were variations in ethnicity, accommodation types, employment status, and state benefits received. Their children’s ages ranged between 2 and 15, and most had a diagnosis of cerebral palsy with varying levels of impairment. Detailed information relating to participant characteristics and recruitment procedures has been reported in the publication outlining the local unmet needs of those caring for children with complex neurodisability (first set of results from the qualitative study, as seen in Figure 1) [24]. These results were triangulated with existing evidence confirming that caregivers of children with complex neurodisability require comprehensive, jargon-free information about their child and available services, support for their own wellbeing, and more joined-up working between health services. This provided much of the rationale for the intervention in this context (ADAPT framework process 2).

Two rounds of semi-structured interviews were conducted with participants, the second round being important for the aims and research question for this paper. Participants were shown a presentation about the “Ubuntu” programme and asked general questions about the content, format, and ways to reach diverse groups. Subsequently, questions were asked about each module; for example, ‘*Is this module important?*’, ‘*What additional information would be important to include?*’ and ‘*What information is not relevant to parents/carers in the UK?*’ Interpreting services were offered for those who required them.

Six healthcare professionals working with children with complex neurodisability from the same community child health centre were interviewed in the same way and asked for their opinions and recommendations. Disciplines included health visitors (*n* = 2), occupational therapist (*n* = 1), paediatrician (*n* = 1), physiotherapist (*n* = 1), and a speech and language therapist (*n* = 1).

Interviews were conducted online, audio-recorded, and transcribed verbatim for analysis. Content analysis using NVivo software was used to summarise the data on recommendations for ‘*content*’, ‘*delivery*’ and ‘*reaching diverse populations*’. In this way, the fourth step of the ADAPT framework process was followed by gathering specific recommendations from local caregivers and healthcare professionals in the community relating to the content, delivery, and how to reach diverse populations.

### 2.3. Exploration of the Context–Intervention Fit

The ADAPT guidance encourages researchers to compare the previous context and the new context, focusing particularly on the similarities between the two [42]. The two settings (Kampala and Kiwoko, Uganda and East London, UK) were compared in site visits by K.P. in discussion with the wider team and through conversations with caregivers and healthcare professionals in each. In exploring the contexts, it would be naive to overlook the political history of coloniality between Britain and Uganda, recognising the historical extraction of resources and exploitation of their people. The traditional direction of knowledge flow was from Britain (the coloniser) to Uganda (the colonised), and it is important to acknowledge the discomfort some may face in the recognition of similarities, such as seeing the NHS as a low-resource health system [36]. Recognising these similarities challenges the notion that people accessing healthcare in Uganda bear no resemblance to those accessing care in the NHS, UK. It presents a new perspective to look beyond the broad-brush strokes of differences between settings, such as ethnicity or economic status, and towards an understanding that resemblance between populations or settings can be nuanced and subtle. For example, it has been made clear that within the context of having a child with a complex neurodisability, regardless of nationality, ethnicity, socio-economic status, or language, parents and caregivers face challenges to their wellbeing [46,47,48,49,50].

### 2.4. Development of the Programme Manual and Programme Theory

We co-created the “Encompass” programme manual through several iterative cycles of adaptation from “Baby Ubuntu”. The ten modules were agreed upon by the adaptation team based on feedback from the qualitative interviews. The parents with lived experience on the adaptation team then advised on the content of the modules to ensure the pictures, texts, and examples provided were appropriate and sensitive to their context and target population. These decisions were explored during workshops through mind mapping, discussion, and paper-based technologies, alongside emails and electronic comments in between workshops/meetings. Further comments and suggestions were collected when the facilitators were trained, and feedback was sought on the manual content. All changes were made by the lead researcher (K.P.) after each workshop and meeting. Plans for delivery were brought to the adaptation team meetings and discussed until consensus was reached. The parents with lived experience advised on key delivery decisions, such as devising a job description for the facilitators and how to reach underserved groups in recruitment. Any differing views between professionals and caregivers were discussed during adaptation, with current evidence and caregivers often given greater power for decision-making. Section A.2 includes a summary table linking feedback to specific adaptation decisions.

The programme theory was developed using a realist methodology to describe the impact of the “Encompass” groups using context-mechanism-outcome (CMO) configurations. This methodology is a theoretically underpinned, pragmatic approach that explores how interventions work, in what circumstances and for whom, rather than only focusing on whether they work or not [51,52]. Realism lies between the positivist lens, where context is seen as a source of bias, and the social constructionist lens, where context is viewed as a foundation of knowledge. For this project, context is therefore described as the interactions between systems, relationships, the way people assign meaning, and the implicit rules that govern how people respond, which all impact the outcomes [53]. It refers to the conditions in the background that would influence the outcomes of the programme. Mechanisms refer to the unseen resources and responses that cause the effects (or outcomes) of the programme through interacting with the context [51]. The mechanisms are based on the behaviour change techniques (BCTs) taxonomy (v1) [54] and the interpersonal change processes from the Mechanisms of Action in Group-based Interventions (MAGI) framework [55]. The BCTs are helpful in providing a common language to describe processes within the programme that aim to change the behaviour of the participants [54]. The MAGI framework provides an additional understanding of how the intricacies of a group-based intervention may influence change within the participants [55]. Outcomes may be intended or unintended based on the interaction between the context and mechanisms [51].

The programme theory was developed with input from the adaptation team, with realist evaluation principles guiding discussion about how caregivers would respond to elements of the “Encompass” programme within their context to achieve the expected outcomes.

## 3. Results

### 3.1. Context–Intervention Fit

Similarities and differences between the two settings (Kampala and Kiwoko, Uganda and East London, UK) are illustrated in Table 1. The differences influenced the adaptation process as they informed decisions about removing content (e.g., related to traditional healers) and adding content (e.g., about the UK schooling system and available equipment for the children). The similarities and differences described in Table 1 relate primarily to the experience of raising a child with a disability and the challenges that accompany this. Many of these experiences are similar; however, there are large structural differences that cannot be ignored, such as the poverty experienced by many in Uganda and the limited access to appropriate healthcare, education, transportation, and housing compared to the UK. Programmes like “Baby Ubuntu” are, therefore, developed within these resource-constrained systems using approaches that are participatory, community-based, low-cost, and often peer-led. It is these types of innovations and approaches developed in LMICs as a response to structural difficulties that may be useful for adoption in HICs.

### 3.2. The Programme Theory

The programme theory produced by the adaptation team for “Encompass” is depicted in a logic model using CMO configurations (Figure 2). Example outcomes are improved wellbeing, empowerment, advocacy, confidence, or knowledge and skills.

The green-filled boxes indicate the context, which ultimately represents the experience of being a caregiver of a child with a complex neurodisability in a society that is not set up for families to participate fully in all areas of life, with the appropriate care and understanding in place. Aspects of the specific setting within the context are described too, such as the NHS facing a workforce crisis and the demographics of the area in which the programme takes place. 

The orange-filled boxes represent the anticipated outcomes based on the interactions between the mechanisms and the context. These outcomes will be explored in a pilot and feasibility study [41].

### 3.3. The “Encompass” Programme

The “Ubuntu” manuals and materials can be copied or adapted to meet local needs; they may be distributed if they are free or not for profit. From its conception, “Ubuntu” has, therefore, been made available with an explicit culture of openness and collaboration attached to it. The “Ubuntu” programme itself has undergone many adaptations because of researchers and organisations sharing resources and ideas, which has then paved the way for projects such as “Encompass” to continue in the spirit of collaboration and adaptation. There were, however, certain core values of the programme that were non-negotiable, such as having a parent with lived experience as one of the facilitators, and ensuring the groups were participatory, including learning from each other rather than module content being ‘taught’. In the current adaptation, two of the modules were merged, with a new module on schools included. However, most of the manual remained the same, with specific activities adapted according to local needs.

### 3.4. Delivery Plan

The co-adapted delivery plan for “Encompass” has been described below in Table 2, using the Template for Intervention Description and Replication (TIDieR) [56] checklist, along with elements from the checklist for reporting of group-based behaviour-change interventions [57]. Illustrative quotes are provided when appropriate from the qualitative study referenced in Figure 1 (C = caregiver, H = healthcare professional). Final decisions were made during co-production workshops with the parent partners with lived experience and during meetings with the wider adaptation team.

Most of the delivery plan remains similar to “Baby Ubuntu”; however, a few key adaptations were made. These included shortening the length of sessions to make it more feasible for families to attend and reducing the number of sessions (through removing and/or combining modules). An online or hybrid option was considered but ultimately decided against based on feedback from the adaptation team.

### 3.5. Content

There was consensus from the caregivers and healthcare professionals in the qualitative study that the “Baby Ubuntu” modules were of relevance to their setting in East London, UK. The modules for “Encompass” are therefore based on the “Baby Ubuntu” modules, with adaptations made based on feedback from the qualitative study referenced in Figure 1. Many of the pictures and activities in the manual were changed to be culturally sensitive to the ethnically diverse urban UK populations.

See Figure 3 below for the “Encompass” modules, followed by Table 3 which provides detailed descriptions of each session and the rationale for adaptations made.

A potential additional module was raised during the qualitative interviews around the topic of caregivers looking after themselves, particularly focusing on mental health. There was consensus, however, that this could be interwoven into each module and caregivers could be signposted to programmes, such as the Healthy Parent Carers [59], where the focus is solely on taking care of their own wellbeing.

### 3.6. Reaching Diverse Populations

Participants in the qualitative study were asked questions around how to reach underserved groups of caregivers, how to account for different languages, and how to consider grouping such a clinically heterogeneous group of children.

A strong recommendation that emerged from most interviews was to consider the atmosphere of the groups. This included suggestions such as ensuring no medical jargon was used throughout the programme, making it a welcoming environment, having refreshments available, having a safe space to share experiences, but being mindful that some may not be ready to speak, and focusing on relationship building between the group members and facilitators.


*“But if they don’t feel comfortable [to share], you shouldn’t pressure them into saying, oh, tell us about your story or tell us your experience if you want. If they don’t then they just enjoy learning about the rest. As long as they’re comfortable.”*
C15


*“I think some people won’t want to share their personal stories. But I guess once they meet up more and they become more comfortable with each other they may be able to start sharing.”*
C3

Group rules or guidelines were recommended to be developed together at the initial group meeting with the assistance of the facilitators. Examples for this included respecting others’ opinions, listening to each other, using mobile phones on silent, and confidentiality (while understanding the limits in relation to disclosure of harm or risk). The above recommendations relate to the best ways of serving diverse groups of caregivers once they have agreed to participate. Examples of how to reach individuals before this stage included making it explicit that interpreting services would be available, advertising where different cultural groups congregate and in public areas and using a variety of paper-based and social media methods.

There was a lack of consensus and uncertainties about grouping together children who may have vastly different clinical presentations and severity of impairments. Some felt that caregivers could gain more from the group if they were grouped with similar children, and healthcare professionals would be able to facilitate practical sessions around positioning with more ease.


*“I wouldn’t want to discuss my son with somebody whose child can’t even speak or needs to be fed. I would as a parent feel slightly awkward or uncomfortable because I don’t understand what they’re going through.”*
C16


*“Diverse groups allow you to learn a little bit, but you can probably learn more if you have children of similar abilities.”*
C15

Others in the study thought that this may make others feel excluded. The decision ended up being a pragmatic one, as there would not be enough participants to group children according to their abilities. During the facilitator training, discussions will be had on how to manage having a group of families who have children with differing abilities, how to create a supportive and inclusive environment, and how to encourage sharing of experiences. This is, however, an uncertainty that should be further explored in the pilot and feasibility study.

Similar views were reported for grouping caregivers according to the language spoken to reduce the amount of time required for various interpreters. However, the same pragmatic decision was made to go ahead with inclusive, diverse groups and explore this issue further in the feasibility and pilot study.

## 4. Discussion

This paper describes the process of adapting the “Baby Ubuntu” programme developed in Uganda to form “Encompass” for the UK. It provides details on the programme theory, delivery plan, and modules of “Encompass”. The programme theory was developed using a realist framework and depicted in a logic model using context–mechanism–outcome configurations. Core elements of the delivery plan were kept, including having a parent with lived experience as a facilitator, and the groups needed to follow a participatory approach. Minor adjustments were made regarding the length and frequency of groups, and it was decided that home visits were not required in this population, as the families received enough support from therapists and other health professionals. It was decided that groups would be held in person, but that online optionality should be explored in the future. Most of the module content remained the same, with minor adjustments made based on local needs. However, one new module was developed, ‘Going to School’. The “Encompass” adaptation is novel in that it is the first time the “Ubuntu” programme has been adapted for a high-income country.

### 4.1. Strengths and Limitations

The strength of the co-adaptation process lies in the participation of those with lived experience being involved in the key decision-making aspects of the “Encompass” programme. Principles from Albert and colleagues [60] describe the importance of efforts shown to shift power and the willingness to share it, to draw from diverse sources, to go to communities rather than expecting them to come to you, and to aim for long-term relationship building rather than short-term. The group of parents with lived experience met in a local community library or online, depending on what was more convenient for them. It was made clear from the beginning that this would not be a one-off project and that the group could continue to meet after this study ended. Parents from this group held greater power in decision-making as their local, lived experience allowed them to provide highly relevant expertise for their context. It is, however, a limitation that only mothers with lived experience were involved in the adaptation team. For future studies relating to this programme, efforts will be made to engage fathers and other under-represented caregivers through community organisations, health professionals, and word of mouth from participants who attended the pilot groups.

Another strength of this study is the systematic approach taken, which involved combining both participatory and theory-based elements in the process. Data from the qualitative study, another key recommendation for adaptation studies [61], were considered alongside the perspectives of the adaptation team to inform decisions around the content, delivery, and how to reach and support diverse and underserved families. The inclusion of key members who were involved in the development and adaptations of “Ubuntu” and “Baby Ubuntu” allowed for valuable knowledge to be shared about previous learnings. The adaptation team did not need to rely solely on publication materials to fully understand the implementation of “Baby Ubuntu”, as queries could be raised with the team themselves.

It is important to note the limitations of the project, particularly when the contexts of Uganda and the UK were contrasted to explore the similarities and differences. Firstly, context is not a ‘thing’ to be researched (a noun), but rather something that ‘happens’ and that researchers ‘do’ (a verb) [62]. Context is complex and dynamic, and interacts with every part of an intervention. The decisions made in this project were guided by observations and discussions at the time, with an understanding that people and places cannot always be simplified and generalised. The descriptions of the similarities and differences between settings described in Table 1 were understood through the lens of the first author (K.P.), a South African, white female currently residing in the UK. KP brings her own experiences of working in low-resource settings in South Africa and the NHS, along with an understanding of her own privilege and how that shapes others’ interactions around her. How the first author made sense of context and its influence on the content of Table 1 might be different for another researcher.

### 4.2. Wider Generalisability

The adaptation followed a similar process to the creation of “Juntos”, an adaptation of the “Ubuntu” programme for families of children with Congenital Zika Syndrome in Brazil [30]. In both the “Juntos” and “Encompass” adaptations, a needs analysis was conducted, and advisory groups were established to support decision-making and the development of the manual. Both programmes developed a theory of change/programme theory to anticipate the mechanisms and impact. Both decided to pilot the groups co-facilitated by an expert parent and a therapist. The “Juntos” manual integrated a new component relating to caregiver emotional wellbeing by providing prompts at the end of each session, and this was incorporated into the “Encompass” manual. Although each programme was adapted for a different context and to support families of children with varying needs, the adaptation and delivery processes followed a similar approach. This suggests that the adaptation model may be transferable to other settings and populations.

The co-adaptation of “Baby Ubuntu” to form the “Encompass” programme is an example of a ‘decolonised healthcare innovation’ [36], as the programme was developed and implemented in LMICs and considered to be a potential frugal innovation for a resource-constrained setting in the UK. ‘Decolonisation’ in global health aims to dismantle ideas about health often created by those with the greatest power [63], thus moving away from a ‘top-down’ approach where the knowledge is most valued from those who have historically held power. The adaptation described in this paper not only values the knowledge created by the Ubuntu Hub teams in LMICs but also the expertise brought by those with lived experience. Using a community-based, participatory, peer-led approach in the co-design, adaptation and implementation of these programmes challenges the dominant discourse that only HICs can generate high-quality research and innovations. It needs to be acknowledged that using the word ‘decolonisation’ as a metaphor can be problematic when it does not explicitly relate to the repatriation of indigenous land and life [64]. We hope, however, that this example may stimulate consideration of other frugal innovations to be co-adapted in child health research, looking beyond the traditional ideas that LMICs are ‘too different’ to consider transferability of findings. Research has demonstrated that interventions, which have been carefully adapted, are more likely to succeed than those adopted or transported between settings without consideration of culture and context. For example, the Africa Clubfoot Training, developed with the University of Oxford, was the first standardised training programme for clubfoot treatment [65,66]. It has since been adapted for use in the UK and other high-income training environments and is accredited by the Royal College of Surgeons in England [67].

### 4.3. Implications and Further Research

The qualitative study conducted with caregivers and healthcare professionals in East London provided preliminary evidence on the theoretical feasibility and acceptability of implementing a co-adapted version of “Baby Ubuntu”, which was further explored during the co-adaptation process of forming “Encompass”. Uncertainties remain, which have implications for further research. Bonell and colleagues [68] recommend piloting implementation when there are uncertainties around the feasibility of delivering an intervention. These implementation uncertainties will, therefore, be explored in the pilot and feasibility study [41]. Whether the implementation of “Encompass” will trigger the intended mechanisms and outcomes, according to the programme theory, will also need to be explored in a larger evaluation of effectiveness.

## 5. Conclusions

“Encompass” is a participatory group programme for caregivers of children with complex neurodisabilities that aims to improve the skills, knowledge, and confidence of caregivers as an example of the implementation of family-centred care. The process of adaptation highlighted the remarkable similarities between the content and delivery plan in both settings (Uganda and the UK). This demonstrates that there is more to ‘context’ than the apparent observable differences (e.g., culture, language, health systems, infrastructure) and that the shared experience of raising a disabled child may surpass these. One of the aims of describing the co-adaptation process is to improve the reporting of adaptation processes, particularly given that this example describes a low-cost innovation developed in low- and middle-income countries being adapted for the first time to a high-income country. Although the results may provide utility to groups interested in the research and implementation of “Encompass”, it is the methods which may be of interest to others and could be replicable in a plethora of contexts.

## Figures and Tables

**Figure 1 ijerph-22-01144-f001:**
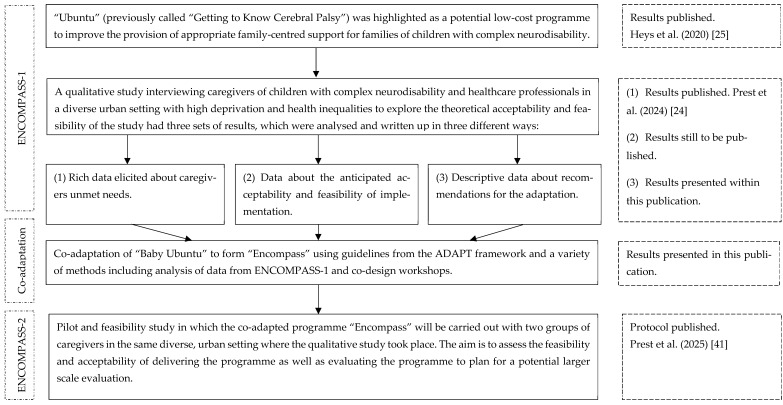
An overview of the different studies involved in ‘ENCOMPASS’ [24,25,41].

**Figure 2 ijerph-22-01144-f002:**
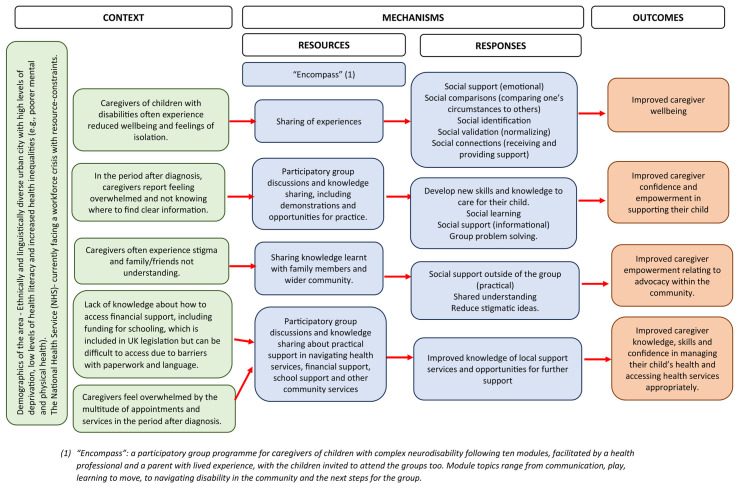
The encompass logic model illustrating the programme theory. The blue-filled boxes indicate the mechanisms of the programme. “Encompass” is briefly described, followed by specific resources offered through the programme. Examples of mechanisms are the sharing of information during the “Encompass” groups (resources) and improved social support, connections, and validation (responses). An example BCT component (and mechanism) is problem-solving within the group, which aims to change the caregiver’s level of confidence and empowerment (the behaviour change/outcome). An example of a MAGI framework component is social validation. Interacting with others in the group in similar situations, caregivers may feel that their own situation is normalised (mechanism), which may have an effect on their wellbeing (an outcome).

**Figure 3 ijerph-22-01144-f003:**
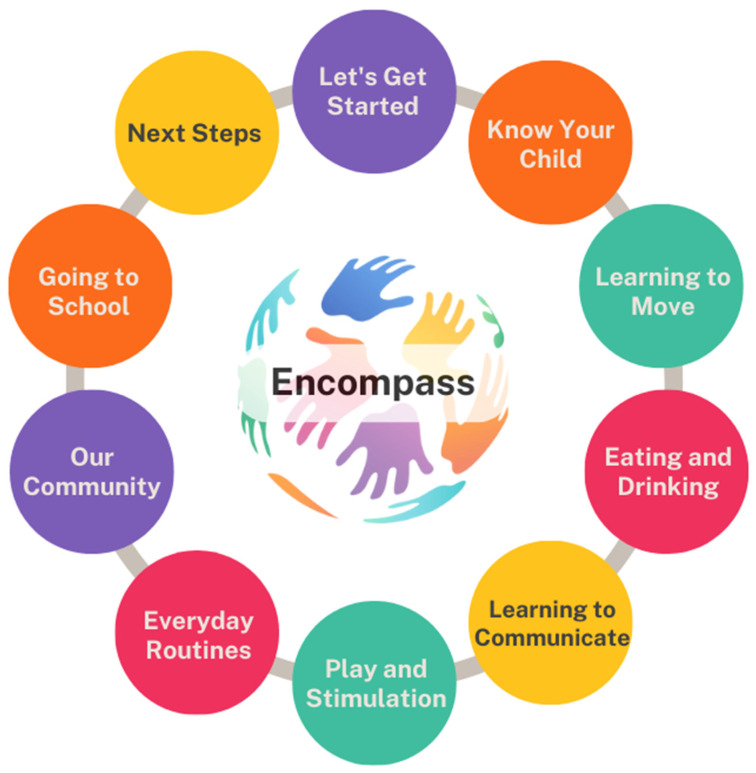
The “Encompass” programme modules [41].

**Table 1 ijerph-22-01144-t001:** Similarities and differences between settings for “Baby Ubuntu” (Kiwoko, Kampala and other parts of Uganda) and “Encompass” (East London, UK).

Similarities	Differences
Having a child with a complex neurodisability results in caregivers feeling isolated and alone. Cultural ideas about the reasons for a disability are prevalent in Kampala or Kiwoko, and some parts of East London (e.g., with older generations). Children with disabilities can be stigmatised in communities in Kampala or Kiwoko (e.g., not wanting to interact with families), and in East London (e.g., not being invited to play with others due to a lack of awareness). In Kampala or Kiwoko, it is not common for fathers to be involved in the care of a disabled child. This is experienced in some parts of East London, but it is not always the case. Hearing stories from other caregivers going through a similar experience helps others to feel less alone across settings. Groups for children with disabilities and their caregivers frequently involve singing, clapping and dancing both in Kampala or Kiwoko, and in East London. Caregivers find it powerful to see a health professional or group facilitator play with their child and show them love. Access to children’s therapy services is limited. While Kampala and Kiwoko have these services (unlike the rest of Uganda), geographical barriers exist. In the NHS in East London, there are barriers based on the way services are set up and whether a child can access a block of therapy. Accessibility is difficult in both Uganda and East London contexts, although to varying degrees, as transport can be particularly difficult in Kampala, with bodas (motorcycles used as taxis) being the most common form of transport. In Kampala or Kiwoko, it is not common to share emotions and vulnerabilities within a group. This is sometimes the case for caregivers in East London, but it can vary. For example, in Bengali communities, it is not easy to share if one is struggling, and this is often kept quiet. In African-Caribbean communities, there can be an over-optimistic sense of ‘you will be okay’.	Children with complex neurodisability are frequently taken to a traditional healer in Kampala or Kiwoko, but not in East London, although some communities may seek religious or spiritual guidance and support. Doctors and nurses generally have more of an understanding about cerebral palsy (CP) and complex neurodisability in East London (e.g., when it comes to malnourishment, feeding or vaccinations); however, some experienced a lack of understanding from their General Practitioner (GP) about feeding difficulties that accompany CP. In Uganda, healthcare workers have been known to blame the parents for their child’s malnutrition or disability. In Kampala or Kiwoko, sometimes other family members became the primary caregiver for the child (e.g., grandmother), which does not happen as frequently in East London. There is limited availability for assistive devices in Uganda, compared to East London, but caregivers in East London describe it being a ‘battle’ to achieve what they need. In Uganda, health professionals are seen as more of ‘experts’ than the caregivers of the child with disability, demonstrating a stronger medical model. Caregivers are unlikely to ask questions or request further support and services. Experiences vary in East London. In Uganda, there are few to no schooling options for children with complex neurodisability unless they have mild impairments and can cope with mainstream school. In East London, there are more supportive school settings. In East London, there are more regular check-ups with specialist neurologists, paediatricians, and orthopaedic consultants than in Uganda. There are more inclusive recreational activities available to families of children with disabilities in East London compared to Uganda, although they are often geared towards other diagnoses, such as Autism or deafness, rather than cerebral palsy.

**Table 2 ijerph-22-01144-t002:** Description of elements of the “Encompass” programme using the checklist for group-based behaviour-change interventions and the TIDieR checklist.

Intervention Element	Description	Comparison with “Baby Ubuntu” Groups
Intervention Design	(Covering Items 7 and 8 on the TIDieR Checklist)	
General setting	There were varied responses about whether to conduct the group sessions online or in a hybrid option; however, the decision was made to conduct the groups in-person due to the practical activities that are part of the programme and to allow social connections to form in-person during the breaks in an informal setting. Online optionality should be explored in the future, particularly as some caregivers explained that they would prefer it. *“Every person is different you know. Like for me it would be better at first video”.* C11 *“I miss the interaction, I miss going [to a previous in-person group], I miss being in that environment. I hate this online thing being at home, I think it really becomes impersonal when you’re online.”* C16 The in-person meeting will be at a local community venue in London, UK (e.g., a library or community hall).	“Baby Ubuntu” groups are always in-person in a community location
Venue characteristics	A large room with space for mats on the floor and chairs to create an informal circular shape for the meeting. Extra space for buggies and wheelchairs, and refreshments during the break. The room should be accessible, along with accessible toilets.	“Baby Ubuntu” groups may be held in a community centre, hospital or even outdoors at times.
Total number of group sessions	10	11
Length of group sessions	Two hours, with a break in the middle for refreshments and to allow caregivers to meet informally.	Timings vary for the groups, and they could take up a whole morning.
Frequency of group sessions	Every two weeks in term-time only.	This varies and could be every week, every two weeks or every month.
Duration of the intervention	Approximately 6 months.	The same, approximately 6 months.
**Intervention content** **(Covering items 3, 4, 6, 11 on the TIDieR checklist)**		
Session content	The sessions cover 10 modules, each with its own topic and accompanying activities and discussion points.	The sessions cover 11 modules, each with its own topic, activities, and discussion points.
Sequencing of sessions	The sessions run from module 1 to 10 sequentially. The “Baby Ubuntu” team developed the sequence of modules as each session builds on the others. This allows for caregivers to practise the skills they have learnt to build their confidence as the groups progress. Caregivers have the opportunity to discuss previous modules with the facilitators if they need to miss one.	The sessions run from module 1 to 11 sequentially.
Participants’ materials	Participants will be provided with an A4 page handout after each session to take home with them. [Regarding handouts] *“When you’re with the child, you wanna have something physically you can hold it like look through it and read it…. Sometimes you’re not going to be like oh, let me take out the laptop and look it up.”* C15	Some of the modules have handouts for caregivers to take home with them after each group.
Activities during the sessions	The groups predominantly follow a participatory learning approach, and although the facilitators have manuals, the activities tend to include some imparting of information, key points for discussion, examples from other settings and activities for the group to practice. *“Yeah, so kind of keeping the structure that [Baby Ubuntu] have. You know how they have an ice breaker activity, then there is like a little teaching moment, and then we have the discussion and so kind of keeping the right balance is in essential.”* H4	The same.
Methods for checking the fidelity of delivery	A fidelity checklist has been developed, which is an adaptation of one that the “Baby Ubuntu” team have used in implementation.	The “Baby Ubuntu” team have a fidelity checklist that they use.
**Participants**		
Group composition	The groups consist of caregivers of children with complex neurodisability under the age of five years. Up to two adults will be invited to attend per child. This does not necessarily need to be the parent, but someone who is involved in the everyday care of the child. Efforts will be made to encourage fathers to attend, if possible, as they tend to be left out of intervention and research activities. *“My brother plays a very active role in our family setting. Although I’m close to my mom and dad, my brother is the one that used to come and help all the time with* [my child].” C16 *“You just predominantly find mothers attending things and that’s just because for my observations anyway, that dad actually struggle a lot more actually emotionally. Their lives aren’t ordinarily impacted as dramatically as moms’ lives are, so sometimes dads don’t feel the urgency to come to these sorts of things. They can still go to work… so I think that’s why sometimes it’s harder to reach dads. So absolutely they should both come, but that’s the problem. Dad is more likely at work.”* C1 Siblings will be invited to the ‘Play and Stimulation’ sessions, and any other carers or family/community members will be invited to the ‘Our Community’ session. The reason for choosing children under 5 to attend is an attempt to capture the group of children and their caregivers who were recently diagnosed, as well as their families moving through this key transition phase. Some families may not be ready to join a group shortly after diagnosis and may need time to first adjust before committing. Others reported that the information would still be relevant 3 years post-diagnosis. [When to invite families] *“The earlier you could get people the better because you just feel so alone, and you want answers.”* C13 *“For me, immediately after I was not quite focused on* [interventions or groups]*. We were still in shock for almost one year.”* C2	“Baby Ubuntu” is a programme for children with developmental disabilities under the age of 3 and their caregivers. Siblings are invited to the ‘play and stimulation’ session, and other carers, family members, and community members are invited to the ‘our community’ session.
Continuity of participants’ group membership	The participants who enrol in a group are expected to continue throughout the modules until the final session, unless they opt out or are unable to continue attending.	The same.
Group size	The groups will aim to have 10 caregivers at a time.	Groups aim to have 8–10 caregivers at a time.
**Facilitators ** **(Covers item 5 on the TIDieR checklist)**		
Number of facilitators	Two facilitators deliver the sessions together.	The same.
Continuity of facilitators’ group assignment	The same two facilitators who begin with a group are expected to complete all modules with the same group, unless circumstances do not allow this.	The same.
Facilitators’ professional background	One facilitator will be an expert parent with lived experience, and the other will be a health professional, in this case, an occupational therapist, or physiotherapist. *“I think* [the facilitator] *needs to definitely be a trained professional, the OT, or whoever it is, but then definitely you would need like a parent/carer facilitator because they’ve just got that relatability and… they can help the professional deliver it in a way that’s going to be better received by the parents, because sometimes you know professionals are passionate about their jobs, and even though they see all of these kids every day, it’s not their life, they will go home. I think it is important to have that relatability.”* C1	The same, although the health professional may have another role besides occupational therapy or physiotherapy. They could be a community healthcare worker.
Facilitators’ personal characteristics	It was agreed that both facilitators should have experience in either navigating (expert parent) or delivering (health professional) healthcare in the local setting. Although there was discussion about whether the facilitators should be able to communicate in frequently spoken languages as an addition to English (e.g., Bengali), consensus was reached that this would not be essential for the role. Facilitators were expected to be able to read and communicate in English in order to deliver the contents of the manual. When recruiting facilitators, there were no preferences with regard to age, gender, ethnicity, or cultural background. In the “Baby Ubuntu” groups, the gender of the facilitators largely depended on the context, with female facilitators being observed to feel more comfortable in supporting the emotional needs of caregivers, and male facilitators having a key role in changing community perspectives.	“Baby Ubuntu” facilitators are required to have enough literacy to be able to read the manual either in English or their local language.
Facilitators’ training in intervention delivery	Facilitators will receive training from a master trainer in “Baby Ubuntu”, which covers both the intervention content, the delivery methods and group facilitation skills. The training manual was adapted with the support of one of the developers of the “Baby Ubuntu” and the subsequent facilitator training manual (RL). Both the parent facilitators and healthcare professional facilitators will be trained together to ensure that power is shared from the beginning. The training is practical and includes group discussions and opportunities to role-play, facilitating activities from the different modules.	The same
Facilitators’ training in group facilitation	See above.	The same
Facilitators’ materials (Item 3 on TIDieR checklist)	Facilitators will be provided with a manual, which includes module 0 that assists them in their preparation for the groups. They will also have materials such as posters and objects to assist with demonstrations and practice during the groups.	The same
Intended facilitation style	The “Encompass” groups, like all previous “Ubuntu” groups, are run with a participatory approach using principles from adult learning theory. The group participants’ experiences are acknowledged as being important knowledge within the group and are built upon during the sessions, along with problem-solving discussions relevant to their own situations. The facilitator imparts their knowledge within this context, but the relationship is more horizontal than vertical.	The same

**Table 3 ijerph-22-01144-t003:** Description of “Encompass” modules and adaptations made from “Baby Ubuntu” modules.

“Encompass” Module	Overview of Module	The Same or Different to the “Baby Ubuntu” Module *?	Adaptations and Rationale
Module 0: Before you begin	Helps facilitators to plan the groups, follow the manual, top tips, common mistakes, how to refer to if there is an issue, and how to monitor and evaluate.	The same	No major changes
Module 1: Let us get started	About the programme, sharing information about complex neurodisability, and personal story sharing.	The same	No major changes
Module 2: Know your child	Explores what their child can do, and what they would like their child to progress to (setting achievable goals) without reinforcing ‘typical developmental milestones’. Exploring different healthcare services.	The same	Although the module has largely stayed the same, parent partners for the project emphasised the need not to reinforce ‘typical developmental milestones’, as this is something that they come across often in medical appointments and interacting with other children in their lives. They felt that this does not need to be reinforced, but rather children should be celebrated for what they can do, and to understand that development will happen differently for all of the children in “Encompass”.
Module 3: Positioning, carrying and learning to move	How to position children who need assistance, and how to assist children to learn to move.	Combined two of the “Baby Ubuntu” modules into one (module 3, positioning and carrying, and module 5, learning to move)	Caregivers as well as healthcare professionals from the qualitative study referenced in Figure 1 agreed that learning to position and carry their child with a complex neurodisability would be an essential component of the “Encompass” programme. Parts of this module would need to be adapted as caregivers may require information about specialist equipment, such as standing frames or seating, which were not as freely available in Uganda for “Baby Ubuntu”. *“I just went by experience and with my son, for example, now I’m struggling because he’s getting quite tall, but it would have helped if I knew how to carry him properly when he was bit younger because I’m having a lot of back pain in general now even when I’m not carrying him and I know that’s because I was carrying him incorrectly”* C16 *“Often parents break their back because of wrong lifting technique and the implication is not on them alone. Even the child that they’re carrying. If they leave the child you know in the wrong position, it could affect so many things”.* H3
Module 4: Eating and drinking	Feeding challenges, practical skills to address these feeding challenges, exploring topics of diet, positioning for feeding, utensils, textures, and kind and sensitive feeding methods.	The same	This module was deemed to be equally relevant to caregivers in London as it was to the participants of “Baby Ubuntu” groups in Uganda. This may include supporting caregivers in knowing when to seek support and acknowledging the stress that accompanies feeding difficulties, as illustrated in the quote below: *“I needed help because every time I had to feed him, we had a lot of problems and I had to keep asking doctors to help me what to do because he was very skinny and he ate a little and even if I wanted, I couldn’t give him you know nutrition. So, I think it should be like that* [caregivers] *can look for help if* [needed].” C11 Additional advice was sought from a local specialist speech and language therapist who provided suggestions for this module, including the introduction of the International Dysphagia Diet Standardisation Initiative (IDDSI) framework [58] to help caregivers understand different textures and thicknesses if their child has swallowing difficulties.
Module 5: Communicating	Explores the importance of communication, how the family can support communication, e.g., engaging, taking turns, making choices, and using alternative forms of communication, and practising communication.	The same	No major changes
Module 6: Play and stimulation	Explores the importance of play for children to develop and learn, early stimulation, using everyday objects in play, inclusion of play in the family and broader community. Siblings were invited to this session.	The same	This module provides an opportunity for caregivers to create simple toys out of everyday objects. In Uganda, toys were made from grasses or plastic bottles during the “Baby Ubuntu” groups. A caregiver in the qualitative study in London made the association that this could still be relevant in their setting. *“She showed me how to make something really simple that was really stimulating and he loved it and it was something that we just made things we had in the kitchen.* C13 The addition of the word ‘stimulation’ assists caregivers to move away from the idea that play has to involve toys and things, creating a broader understanding that stimulating activities can promote play or playfulness. *“And I like early stimulation. I think that’s really important because looking at it, people think of play as almost this specific thing, but it is the early stimulation that you can do leads into play”* H4
Module 7: Everyday Routines	How to support children in their everyday routines, for example, sleep, dressing, self-care, and managing travel.	The same (although the module title changed from ‘Everyday Activities’)	The title of ‘Everyday Activities’ was altered slightly to ‘Everyday Routines’ so that the routines of the child could be considered throughout their day and night. With the implementation of “Baby Ubuntu”, this module provided an opportunity to practise many of the skills learnt during previous modules around positioning, eating, drinking, and communicating. Additional self-care activities (e.g., dressing) were noted as important by caregivers in the qualitative study. They also brought up sleep difficulties, including pain, breathing difficulties and leg spasms, which aim to be included in the discussion points.
Module 8: Our Community	Identifying ‘our community’. Discussion about common barriers to inclusion and knowing the rights of people with disabilities.	Combined two of the “Baby Ubuntu” modules into one (module 9, togetherness and belonging, and module 10, our community)	Both caregivers and healthcare professionals from the qualitative study highlighted the importance of understanding local charity support in the ‘Our Community’ module. They reported that the “Encompass” groups could be valuable in signposting caregivers to different supports and that the group may be able to learn from each other. *“There are various charities out there that help with fund raising for different things like kids shoes. ‘Cause I have to get two different sizes and it’s a nightmare”* C13 *“There’s always lots of little groups going on that people don’t know about”* H4 Although signposting was considered a helpful aspect of the above module, other caregivers emphasised the importance of being proactive and empowering parents to seek out information, as local support offers tend to change. This relates to one of the anticipated outcomes of the programme theory, which is to improve caregivers’ health literacy and activation by improving their skills, confidence, and knowledge in seeking out, accessing, and interacting with various services for their own or their child’s health.
Module 9: Going to school	Knowing where to go to seek advice and support for schooling, understanding some of the wording used in schools, e.g., EHCPs, SEND, and sharing experiences and common concerns in sending their children to school.	New module developed for “Encompass”	This module was a new recommendation, which was not included in any previous adaptations of the programme. The additional module within the “Encompass” programme relates more to understanding the policies and systems in the UK that are in place to support the inclusion of children with disabilities within schools. *“Education needs to definitely have its own thing because there are changes between nursery school and then there were changes from primary school to secondary school. And funding? One to one? Should your child go to mainstream school? Should the SENCo help you know? What is an EHCP plan? Writing it during your application that took me flipping ages.”* C1 *“So I sent him to a normal nursery even though we were receiving help from like OT. But the SENCo didn’t know about it till late last year and now I’ve got to move him to a special needs nursery at the end of year. Whereas if I was told from the get-go, none of that would have happened. So now I’ve got uproot him to somewhere he’s been for a year and a half.”* C3
Module 10: Next steps	Making plans to continue the group, finding other communities, summarising learnings, and sharing significant changes.	The same	No major changes

* Even if the module was kept the same as “Baby Ubuntu”, all modules underwent minor changes to ensure activities, pictures, and scenarios were culturally appropriate for the local context.

## Data Availability

The original contributions presented in this study are included in the article material. Further inquiries can be directed to the corresponding author.

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
