# Peer review of "Adapting a Participatory Group Programme for Caregivers of Children with Complex Neurodisability from Low-, Middle-Income Countries to a High-Income Setting: Moving from “Baby Ubuntu” to “Encompass”"

_ijerph, 2025, doi:10.3390/ijerph22071144_

Round 1

Reviewer 1 Report

Comments and Suggestions for Authors

The manuscript presents a valuable adaptation of the "Baby Ubuntu" programme for high-income settings, demonstrating methodological rigor and stakeholder engagement. Below are key comments for the authors to address:

  • Provide a detailed breakdown of how each ADAPT phase (e.g., assessment, decision-making, adaptation) was operationalised. Presenting this in a table or appendix would enhance clarity for readers.

  • Expand on the rationale for adapting an LMIC-derived programme to the UK context, citing examples such as the NHS’s adoption of frugal technologies.

  • Provide more detail on how qualitative insights (from caregivers/professionals) were triangulated with existing evidence or literature to support methodological rigour.

  • The adaptation process currently lacks implementation data. If piloting or uptake results are not yet available, explicitly state this and indicate when they will be shared.

  • Include a summary table linking stakeholder feedback to specific adaptation decisions to enhance transparency and replicability.

  • Acknowledge the limitation of involving only mothers in the co-creation team. Discuss actions taken or future plans to engage fathers, non-binary caregivers, and parents of younger children to ensure broader inclusivity.

  • Clarify how caregivers were recruited, how neuro-disability was defined, and how underserved or marginalised families were identified and reached.

  • Explain how differing views from caregivers and professionals were managed during co-creation. For example, clarify whether decisions were based on consensus, evidence, or contextual feasibility.

  • Discuss how power imbalances between professionals and caregivers in co-creation workshops were identified and mitigated.

  • Clarify how the adapted programme addresses the needs of ethnically diverse urban UK populations (e.g., multilingual resources, culturally sensitive materials).

  • Some comparisons between Uganda and East London feel overstated. Provide a more nuanced discussion of structural differences to contextualise the reverse innovation claim.

  • Improve the discussion of decolonisation by clearly defining the term, referencing key frameworks (e.g., Tuck & Yang, 2012), and explaining how programme features (e.g., peer-led facilitation) align with decolonial principles.

  • Simplify dense theoretical content by adding brief explanations or real-world examples (e.g., for “family-centred care” and “reverse innovation”) to aid interdisciplinary readership.

  • Provide detail on how the programme will be embedded within existing NHS services—such as pathways for staff training, funding mechanisms, and sustainability strategies.

  • Revise Figure 1 to ensure readability (clear text, consistent formatting) so that it better supports the main narrative.

  • Confirm whether ethical approval was obtained, especially in light of the involvement of potentially vulnerable participants.

  • Briefly describe how emotional and psychological risks in caregiver workshops were identified and mitigated.

  • Simplify dense sections, particularly in the Introduction. Use subheadings or bullet points to enhance clarity.

  • Define key terms like “family-centred care” early in the manuscript to support comprehension among readers from different disciplines.

  • Provide a clear comparison between the original Baby Ubuntu programme and the UK-adapted version. A side-by-side table of module topics and delivery approaches would enhance clarity.

  • If available, include early feedback from the ongoing pilot study (e.g., caregiver engagement, relevance of content) to support feasibility claims.

  • Indicate in the abstract that this is a process-focused paper to set accurate expectations for readers.

Author Response

Thank you for reviewing the manuscript, please find responses attached. 

Reviewer 2 Report

Comments and Suggestions for Authors

Thank you for the opportunity to review the manuscript ‘Adapting a participatory group programme for caregivers of children with complex neuro-disability from low-, middle-income countries to a high-income setting: moving from “baby ubuntu” to “encompass”’. I thoroughly enjoyed reading this manuscript. The area of research is vital, not only in terms of interventions for informal supporters and children with disability, but in its decolonising approach and sensitivity. It is also theoretically and methodologically sound. I commend the research team, the lived experience advisors, and the healthcare professionals involved.

Additional detailed comment is provided below. There does appear to be some formatting issues with the paper, particularly the tables and figures. I am not sure if this was a result of uploading the document.

Abstract

Line 32: I think it should be ‘rationale’

Lines 39-41: I think this is a bit repetitive. I would be very keen to have a sentence or two instead adding detail to the ‘decolonising healthcare innovation’ finding.

Also please make it clear in the abstract that the study focuses on cerebral palsy.

1. Introduction

The first paragraph requires some rewording. For example, perhaps it would be more appropriate to highlight the stress of multiple appointments. Additionally, perhaps rather than suggesting there are a disproportionate number of services involved, it may be useful to tease this out a little more – for example are multiple services involved which do not necessarily need to be because of the way the service systems operate?

Third paragraph: Please provide additional detail about the East London study e.g., was it caregivers of children with CP?

Figure 1: Very useful. I am not sure if the formatting was affected in the uploading of the document. Some of the text has been obscured.

2. Methods

2.1 Adaptation team

This paragraph is in effect one sentence which currently is difficult to read. Please consider breaking this up into multiple sentences.

2.2 Local perspectives

Line 188: Please delete ‘the’ before East London.

2.3 Exploration of the context-intervention fit

An important, thoughtful, and thought-provoking paragraph.

It would be useful to have a little more demographic detail about East London here or in this section rather than waiting until the detail in Figure 2.

3. Results

Table 1: Important information which sits well in a table. The current formatting makes it a little difficult to read. Perhaps consider a slightly different presentation with more space between the dot points?

Table 2: There appear to be font changes throughout the table. Also, there appears to be additional spacing between some words.

The use of illustrative quotes was very useful.

What happens to caregivers who miss a session given they are sequential?

 Table 3: Please check the punctuation in this table as it is not consistent. Also Encompass is in all capitals in module 8.

3.6 Reaching diverse populations

You discuss the differences in opinion around grouping parent/caregivers with children with different/the same presenting level of disability. While you mention the decision was a pragmatic one, given the differences in views about this, it would be useful to include what strategies will be used to manage this in the sessions.

4. Discussion

I think it would be useful to tease out the similarities/differences between this adaptation attempt and that of the Juntos program. It provides important context and implications which are a little underdeveloped in the discussion at the moment. If word count is a problem, you could perhaps simplify some of the detail in the Strengths and Limitations section.

Author Response

Thank you for reviewing this manuscript, please find responses in the attachment. 

Reviewer 3 Report

Comments and Suggestions for Authors

I am grateful for the opportunity to become acquainted with such an important and timely topic, which the authors have presented in a thorough and engaging manner.

I kindly ask you to consider the following suggestions for improvement:

  1. Structuring the abstract so that it follows a more typical and clear format (e.g., with a distinct division into: aim, methods, results, and conclusions).
  2. Improving the figure on page 4, as editorial errors have caused some text fields to be partially obscured or unreadable.
  3. Rewriting the methods section, possibly by presenting the individual steps in a graphical or tabular form — which may significantly enhance its clarity and readability.

Thank you for the opportunity to review the manuscript.
Sincerely,
Reviewer

Author Response

Thank you for reviewing the manuscript, please see attachment for our responses. 
